# Association between Dietary Patterns and Low HDL-C among Community-Dwelling Elders in North China

**DOI:** 10.3390/nu13103308

**Published:** 2021-09-22

**Authors:** Pengkun Song, Qingqing Man, Yuqian Li, Shanshan Jia, Dongmei Yu, Jian Zhang, Gangqiang Ding

**Affiliations:** National Institute for Nutrition and Health, Chinese Center for Disease Control and Prevention, 29 Nanwei Road, Xicheng District, Beijing 100050, China; songpk@ninh.chinacdc.cn (P.S.); manqq@ninh.chinacdc.cn (Q.M.); liyq@ninh.chinacdc.cn (Y.L.); jiass@ninh.chinacdc.cn (S.J.); yudm@ninh.chinacdc.cn (D.Y.)

**Keywords:** dietary pattern, low HDL-C, elders, cross-sectional study, North China

## Abstract

We aimed to investigate the association between dietary patterns and low HDL-C among the elderly population living in North China. The data were from a national cross-sectional survey conducted in 2015. General information in terms of living habits, health status, and food intake using 24 h dietary recall for three consecutive days was procured, and the weight of edible oil and condiments recorded. Anthropometric index, blood pressure, and fasting serum lipids were measured using standard methods. Dietary patterns were derived from food categories by exploratory factor analysis, and multivariate logistic regression was used to estimate the odds ratios of low HDL-C across quartiles of dietary patterns. Among 3387 elderly participants, 21.9% had low HDL-C levels. After adjusting for potential confounding factors, participants with highest score versus lowest score in the balanced dietary pattern had a decreased risk of low HDL-C (OR = 0.38, 95% CI: 0.16–0.88, *p* for trend = 0.013) in the group with a BMI of 27.1 kg/m^2^ and above. Compared to the lowest quartile, there was a statistically significant negative association between the highest scores of the Western dietary pattern and low HDL-C (OR = 0.37, 95% CI: 0.17–0.82, *p* for trend = 0.018) in the group with a BMI of 21.6–24.8 kg/m^2^. However, greater adherence to a thrifty dietary pattern (highest quartiles vs. lowest quartiles) was associated with increased risk of low HDL-C (OR = 3.31, 95% CI: 1.05–10.40, *p* for trend = 0.044), especially in the subgroup with a BMI of 21.6 kg/m^2^ and below. The study revealed that it is urgent to develop district-specific dietary improvement plans for dyslipidemia based on the nutritional status of the elderly population in North China.

## 1. Introduction

Dyslipidemia is one of the well-established risk factors for atherosclerosis and heart disease and is determined to account for more than half of coronary artery disease (CAD) cases worldwide [1,2]. In contrast with other Western countries, the most common phenotype of dyslipidemia in China and other Asian countries is low levels of high-density lipoprotein cholesterol (HDL-C) and elevated levels of triglycerides (TG) [3,4], which predisposes a person to premature CAD and diabetes. As far as we know, HDL-C plays an anti-atherosclerosis role through the reverse transport of cholesterol. Previous observational studies have found that reduced HDL-C levels are an independent risk factor for atherosclerotic cardiovascular disease events [5], with studies confirming HDL-C as a strong independent risk predictor of CHD [6]. Observational studies indicate that cardiovascular risk decreases by nearly 2% to 3% per 1 mg/dL increase in HDL-C [7,8].

HDL-C is one of the most complex lipoproteins in the blood, and its concentration is influenced by endogenous and exogenous factors. In fact, lifestyle and environmental factors, including physical activity, diet, and smoking, have been identified to be associated with serum HDL-C concentrations and to be relevant modulators of the CHD biomarker [9]. For decades, studies have reported an association between food items or specific nutrients and HDL-C level. For instance, fish and n-3 fatty acids, especially DHA and EPA, have a modest effect on LDL-C and HDL-C. Harris’s study estimated that the consumption of 4 g/d of EPA and DHA increased HDL-C by 1%–3% [10]. Moreover, a meta-analysis of 60 controlled studies showed that replacing 10% of dietary energy from carbohydrates with saturated fats, monounsaturated fats, or polyunsaturated fats results in increases in HDL-C of 4.7, 3.4, and 2.8 mg/dL, respectively [11]. 

Dietary patterns reflect the overall dietary structure and characteristics by summarizing various foods into a set of markers, rather than focusing on a single diet substance or nutrient, which is of more significance for improving residents’ diets [12]. Association between dietary patterns and HDL-C is also reported. A systematic review of 50 studies summarized the effect of a Mediterranean diet on multiple components of metabolic syndrome, and identified that HDL-C levels improved with adherence to the Mediterranean diet by increasing by 1.17 mg/dL in 29 clinical trials [13]. However, the evidence was based mostly on studies conducted Western countries, with few studies investigating the association between dietary patterns and HDL-C levels among the Chinese, especially the Chinese elderly population.

With the development of the economy and acceleration of urbanization, the dietary structure of the Chinese population has changed dramatically [14], with the changes affecting the blood lipid profiles. Reports show that TC, TG, and LDL-C do not linearly increase with age, but there is a trend for the HDL-C levels to increase with age [15]. However, the ability of blood lipids to predict coronary artery events decreases with age and there is a lack of direct correlation between coronary artery events and lipid levels, especially in older women. Reverse epidemiology was detected in a variety of chronic diseases, but the phenomenon remained controversial and the underlying causes were poorly understood. From this point of view, dietary nutritional requirements among the elderly will be different from others, especially those with various diseases. Meanwhile, according to the differences in living habits and dietary patterns across specific districts in China, this study was designed around the elderly population living in the northern geographic region of China, which is a typical district including two metropolises, three provinces, and one autonomous region. This study aimed to (1) assess the dietary patterns and health conditions among elderly people living in North China and (2) demonstrate the association between low HDL-C and dietary patterns to provide a scientific basis for promoting healthy aging in the people of this region of China. 

## 2. Materials and Methods

### 2.1. Data Source and Sampling Methods

The data were from China Adults Chronic Diseases and Nutrition Surveillance in 2015 (CACDNS2015), a nationwide cross-sectional survey conducted in 31 provinces, autonomous regions, and municipalities directly under the central government, throughout China (except Taiwan, Hong Kong, and Macao) [16]. A stratified multistage cluster random sampling method was used to identify the participants in 302 survey sites, based on 605 monitoring sites of the disease surveillance points system. In stage 1, three townships were randomly selected in each surveillance site using systematic sampling based on population size. In stage 2, two administrative villages or neighborhood committees were selected randomly in each township also using systematic sampling based on population size. In stage 3, using a simple random sampling method across several village or resident groups, 60 households were selected per group from each administrative village or neighborhood committee. In stage 4, using a simple random sampling method, 45 households were selected from each village or resident group, in which 20 were dietary households and 25 were non-dietary households. Finally, 270 households and 612 regular inhabitants (adults ≥ 18 years old) were chosen for the study. The surveillance was approved by the Ethical Committee of the Chinese Center for Disease Control and Prevention (No. 201519-B). All the participants provided written informed consent. Due to the representative nature of provinces and regions in this surveillance, participants aged 60 years and above living in Beijing municipality, Tianjin municipality, Hebei province, Shandong province, Shanxi province, and the Inner Mongolia autonomous region (areas classified as North China geographically) were analyzed in our study. 

### 2.2. Data Collection and Dietary Pattern Assessment

A standard questionnaire was used along with face-to-face interviews of participants at their homes by trained investigators. The questionnaire included the participants’ general demographic information, such as information on age, education level, marital status, income, health status, and life behaviors related to health, such as smoking and alcohol consumption. Information on the intensity of, and time spent on, physical activities, such as farm work, housework, transport-related physical activity, exercise, and recreation in a typical week, was also collected.

Dietary information was obtained by 24 h recall for three consecutive days. It included information on all the food consumed for breakfast, lunch, and dinner; all liquids taken, such as soft beverages and wine; and any snacks or other foods consumed at or away from home, except for edible oils, water or energy-free water, soups, and flavorings. Edible oil and condiments such as salt, sauce, and other flavorings in the household used during these three consecutive days by food weight were record during the investigation. For those who were unable to provide diet information themselves, family members who took care of them or prepared meals for them completed the questionnaire. We used China national food composition 2004 [17] and 2009 [18] editions to redefine food categories based on similar types of food and nutrient profiles. Because some food items were consumed by fewer than 5% of the participants, for dietary pattern analysis, the foods were further divided into the following 18 categories: rice and rice products, wheat and wheat products, other cereals (millets, sorghum, maize, buckwheat, etc.), tubers (sweet potatoes, Chinese yam, taro, potatoes, etc.), dried beans (mung beans, red beans, kidney beans, peas, cowpea, etc.), soybeans and soybean products (soybeans, black beans, green beans, tofu, bean curd, bean curd cake, soybean milk, etc.), fresh vegetables, fresh fruits, nuts, pork, other red meats (beef, mutton, horsemeat, etc.), red meat offal (liver, belly, lungs, etc.), poultry (chicken, duck, goose, etc.), milk and dairy products (milk, milk powder, cheese, yogurt, etc.), eggs, aquatic products (fish, shrimp, and shellfish), cakes and desserts, and candy and starches.

Exploratory factor analysis and principal component analysis were used to identify the dietary patterns involving the 18 food groups. Bartlett’s test of sphericity and the Kaiser-Meyer-Olkin (KMO) measure of sampling adequacy were used to verify the adequacy of the correlation matrices with the data. Factor analysis was then used to extract the major components. A component was retained on the basis of the following criteria: eigenvalue > 1.25, the scree plot, factor interpretability, and the variance explained > 4.5%. The factors were made independent of each other by a varimax (orthogonal) rotation and to achieve a simpler structure with a higher interpretability. Usually, food categories with absolute factor loading ≥ 0.20 were determined to be strongly related to the identified factors. Factor scores were divided into four quartiles based on their distribution in each stratum. Then the mean and the standard deviation (SD) across the four quartiles were used to present the average consumption in each quartile of each food item for each dietary pattern.

### 2.3. Anthropometric Data and Biomarker

Anthropometric measurements included height, weight, and waist circumference. Trained field health workers measured height and weight based on the standardized protocols of the World Health Organization. Weight was measured without shoes and in light clothing to the nearest 0.1 kg. Height was determined to the nearest 0.1 cm without shoes. Waist circumference was measured to the nearest 0.1 cm. The body mass index (BMI) was calculated as weight (kg) divided by height (m) squared. Based on the criteria recommended by the Working Group on Obesity in China, the BMI was divided into four levels: underweight: BMI < 18.5 kg/m^2^; normal: BMI 18.5–23.9 kg/m^2^; overweight: BMI 24.0–27.9 kg/m^2^; obese: BMI ≥ 28.0 kg/m^2^. Qualified health workers tested blood pressure three times using an Omron HBP 1300 electronic sphygmomanometer. 

Fasting venous samples were collected by qualified nurses by venipuncture from the antecubital vein before breakfast in the local site, and centrifuged at 1500× *g* for 15 min after being left standing for 30 to 60 min. All centrifuged serum samples were transported to a laboratory and stored at −80 °C. The blood collection procedure, processing, and determination were according to quality control standards. Serum lipids, such as total cholesterol (TC), TG, HDL-C, and low-density lipoprotein cholesterol (LDL-C), were tested using a Hitachi automatic biochemical analyzer, with reagents from the central laboratory of the National Institute for Nutrition and Health, China CDC. According to the Chinese Adult Dyslipidemia Prevention Guide (2016 revised edition), serum high-density lipoprotein cholesterol concentrations lower than 1.04 mmol/L (40 mg/dl) were considered to be low HDL-C [19].

### 2.4. Inclusion and Exclusion Criteria

In the surveillance, participants aged 60 years and above who were in favorable condition and able to walk freely to the survey location and answer most questions fluently, were included. At the survey location, anthropometric parameters were measured, and venous blood collected. Where basic demographic information was missing, severe chyle blood samples were excluded following unified data cleansing rules. Samples with missing values were eliminated, and illogical data, such as energy intakes lower than 800 kcal or higher than 4000 kcal in males and less than 500 kcal and more than 3500 kcal in females, were deleted. 

### 2.5. Statistical Analyses

All survey data were collected using specialized software, and data were analyzed using Statistical Analysis System (SAS) for Windows V9.4 (SAS Institute, Cary, NC, USA). The difference among participants’ general characteristics between two HDL-C statuses was compared using a Student’s *t*-test and chi-squared test for continuous variables and categorical variables, respectively. Pearson correlation analysis was used to test significant variables such as age, BMI, waist circumference, blood lipids, and physical activity time. Three dietary patterns were derived by factor analysis using the principal component method. Linear trend regression analysis was conducted using the general linear model and the Cochran-Mantel-Haenszel test for continuous and categorical variables, respectively, to identify the relationship between characteristics and factor scores of each dietary pattern. Linear regression analysis was also used to investigate any significant differences between dietary patterns across the four quartiles of food intake.

We used multivariate logistic regression analysis to evaluate the association between dietary pattern scores and low HDL-C by adjusting for the potential confounding factors of age (60 y, 65 y, 70 y, 75 y, 80 y), education (low, median, high), living area (urban/rural), nationality (Han/other), marital status (having a partner/other status), smoking status (yes/no), drinking status (yes/no), moderate and high-intensity physical activity time (<150 min/week and >=150 min/week), annual household income (low, median, high), BMI (underweight/normal weight/overweight/obesity), and total energy intake (continuous). A stratified logistic regression analysis was also used to further adjust the effect of BMI for its significant confounding. A bilateral *p*-value < 0.05 was considered statistically significant. 

## 3. Results

The basic characteristics of the older participants are shown in Table 1. Of the 3387 participants, 21.9% (*n* = 741) had low HDL-C (25.0% in males and 18.7% in females). With an increase in the education level, the proportion of elderly people with low HDL-C increased significantly, from 20.7% to 26.1% (*p* = 0.012). There was also an increase in elderly people with low HDL-C in proportion with an increase in income (*p* = 0.022). The prevalence of low HDL-C across the four BMI subgroups was 30.3%, 15.5%, 25.3%, and 29.5%, respectively. The average means of BMI, waist circumference, and triglyceride levels were significantly higher in low HDL-C elders than in normal participants (*p* < 0.001), while total cholesterol and LDL-C levels were lower in the low HDL-C group than in the normal HDL-C group. Significant differences were also found between HDL-C status and gender, nationality, drinking status and physical activity (*p* < 0.05).

Factor solutions seen in the whole older population, for male and female elders, were similar, and their dietary patterns obtained by factor analysis of all the participants are presented in Table 2. The first pattern extracted by a factor solution, with an eigenvalue of 1.83, loaded heavily with vegetables, pork, eggs, fruits, nuts, aquatic products, and rice, was named the balanced pattern. The second pattern (eigenvalue = 1.38) was named the Western pattern and was characterized by a higher intake of milk and dairy products, other red meats, cakes and desserts, fruits, poultry, nuts, dried beans, and rice, and a lower intake of wheat and other cereals and vegetables. The last food pattern was named the thrifty pattern and was characterized by a higher intake of other cereals, tubers, soybean products, candy and starch, and vegetables, with a lower egg intake. These three factors could explain 24.9% of the variance in the food intake.

The characteristics of the older participants in North China in quartiles of three dietary patterns are provided in Table 3. Those with higher scores for balanced and thrifty patterns were younger, while those with higher scores for the Western pattern were much older. Females tended to have a higher score for the Western pattern but a lower score for the balanced and thrifty patterns. Participants with higher scores for the balanced and Western patterns were both more likely to live in urban areas and have better income and education status. Participants with higher scores for the thrifty pattern seemed to be current smokers, while drinkers had higher scores for the balanced pattern. As for physical activity, participants with high scores in the three patterns were all most likely to have adequate activity. For the health-related index, the participants in the top quartile of the balanced and Western dietary patterns had a higher BMI than those in the lowest quartiles (*p* for trend < 0.01). In contrast, participants with higher scores for the thrifty pattern had lower BMIs (*p* for trend = 0.017). Similarly, elders with higher scores for the balanced pattern tended to have larger waist circumferences.

In addition, elders in the top quartile of the Western pattern had higher TC, TG, and LDL-C levels than those in the lowest quartiles (*p* for trend < 0.05). However, participants with the highest scores for the thrifty pattern had low TC, HDL-C, and LDL-C levels (*p* for trend < 0.05). Participants with high scores for the balanced pattern had significantly higher HDL-C levels than those in the lowest quartile (*p* for trend < 0.01). No liner significant difference was found in HDL-C levels across the four quartiles of the Western pattern (*p* for trend = 0.073). Moreover, Pearson correlation analysis among significant variables showed that BMI is a strong and significant indicator of the levels of blood lipids and other variables (Appendix A). Thus, a stratified analysis was essential to adjust for the relationship between different dietary patterns and low HDL-C. 

Table 4 displays the main foods (absolute loadings > 0.2), energy and macro-nutrient intake in the quartiles of the three dietary patterns. In the balanced pattern, participants in Q4 consumed significantly higher amounts of rice, fresh vegetables, fresh fruits, nuts, pork, eggs, and aquatic products when compared with Q1 (*p* for trend < 0.001). For the Western pattern, participants in Q4 consumed significantly higher amounts of rice and products, dried beans, fresh fruits, nuts, other red meats, poultry, milk and dairy products, and cakes and desserts (*p* for trend < 0.001) and significantly lower amounts of wheat products and fresh vegetables (*p* for trend < 0.001). As for the thrifty pattern, the participants in Q4 consumed higher amounts of other cereals, tubers, soybeans, fresh vegetables, candy and starches while consuming fewer eggs (*p* for trend < 0.001).Compared with Q1, the energy and carbohydrate intake was significantly higher in Q4 among participants following the balanced pattern and the thrifty pattern (*p* for trend < 0.001). For those following the Western pattern, the energy and carbohydrate intake was higher in Q1 than in Q4 (*p* for trend < 0.001). 

Table 5, Table 6, Table 7 and Table 8 showed the association between the quartiles of the three dietary patterns and low HDL-C by classifying the quartiles of BMI. The BMI quartiles among the elderly participants were 21.6, 24.8, and 27.1 kg/m^2^, respectively. After adjusting for general socioeconomic characteristics, lifestyles, waist circumference, blood pressure, and energy intakes in the final models, the risk of low HDL-C increased with the thrifty pattern scores in the participants in the lowest quartile of BMI (a BMI of 21.6 kg/m^2^). The odds ratio of low HDL-C in the final model was 3.31 (95% CI: 1.05–10.40) in the Q4 score of the thrifty pattern compared with the Q1 score (*p* for trend = 0.044). However, in the BMI subgroup of 21.6–24.8 kg/m^2^, there was a significant negative association between the highest scores of the Western pattern and low HDL-C. The OR value in the Q4 score was 0.37 (95% CI: 0.17–0.82) compared with the Q1 score in the final model (*p* for trend = 0.018). As for the highest quartile of the BMI subgroup with 27.1 kg/m^2^ and over, the balanced pattern was significantly negatively associated with low HDL-C, with an odds ratio of 0.38 (95% CI: 0.16–0.88) in the highest score of the balanced pattern in the final logistic regression model (*p* for trend = 0.013). Nevertheless, there was an interaction between the three dietary patterns, that is, with increased intake of the other two dietary patterns, the negative association between the balanced pattern and low HDL-C and the positive association between the thrifty dietary pattern and the low HDL-C increased. 

## 4. Discussion

In this study, three dietary patterns of participants aged 60 years and above living in North China were identified: balanced, Western, and thrifty. The balanced pattern, characterized by a higher intake of vegetables, pork, eggs, fruits, nuts, aquatic products, and rice, was positively related to HDL-C levels (the HDL-C levels were high), especially in individuals with higher BMI levels, of 27.1 kg/m^2^ and over. We found a significant positive association between the highest scores of the Western pattern and HDL-C levels (the levels were good) in the individuals with adequate BMI levels (21.6–24.8 kg/m^2^). Contrarily, there was a strong negative association between the thrifty pattern, characterized by a higher intake of other cereals, tubers, soybean products, candy and starch, and vegetables and lower intake of eggs, and HDL-C levels (the HDL-C levels were low) in the group with a BMI lower than 21.6 kg/m^2^. These associations were similarly identified after adjusting all potential factors among the elderly population.

Blood total cholesterol, LDL-C, and HDL-C levels have been correlated via adequate research on LDL-C and total cholesterol over the years. However, due to the difficulty of improving the blood HDL levels with medicine and other measures, low HDL-C has been relatively less studied than hypercholesterolemia and high LDL-C in European countries. Nevertheless, this being a common type of dyslipidemia in Asia, it was necessary to study the dietary effect on HDL-C levels in the Asian population. One study on the dyslipidemia pattern in Korean adults found that the TC/LDL-C pattern is related to high dietary fat and cholesterol intake, but that the TG/HDL-C pattern is associated with low intake of milk and dairy products [20]. MediterrAsian diet products, such as red yeast rice, bergamot, artichoke, and virgin olive oil, have promising effects on increasing the HDL-C serum levels [21]. Thus, we analyzed the association between diet and low HDL-C in elderly Chinese people.

The traditional Chinese diet includes high amounts of cereals and vegetables and low amounts of animal-based foods [22]. This dietary pattern is inversely associated with weight gain and hypertension [23] and consistent with the findings of a cross-sectional study of CNHS [24]. This finding is also supported by other studies conducted in Thailand [25] and Korea [26]. However, a balanced dietary pattern was extracted by factor analysis among the elders in North China in our study. The interesting findings revealed that the consumption of high amounts of plant-based foods and animal-sourced foods seems to benefit the lipid mechanism in the elderly population. This balanced dietary pattern includes not only a higher intake of rice, fruits, nuts, and vegetables but also a higher intake of eggs and aquatic products. In fact, this pattern increased the level of HDL-C while not influencing the level of other components of lipids in our research. For elderly people, fish intake at least twice per week is especially advised because it supplies high-quality protein and n-3 poly-unsaturated fatty acids, which help to increase the HDL-C levels [27]. Although eggs are a major source of dietary cholesterol, they are also a good source of high-quality protein, unsaturated fats, and some vitamins and minerals [28]. Moreover, there is no evidence of any association between egg intake and dyslipidemia [29], whereas eggs can improve the lipid profile [30]. Furthermore, fruits, nuts, and vegetables might reduce the risk of CVD beyond the effects of the lipid mechanism, which might be attributed to some nutrients, soluble fiber, polyphenols, and vitamins [31]. Studies have also shown that a nutritious and balanced diet is important for promoting and maintaining healthy aging, and many studies have demonstrated that improvements in diet contribute considerable health benefits for elderly people [32,33,34]. These findings indicate the role of a balanced diet in health promotion, and that a balanced diet is recommended in the dietary guidelines of China and other countries.

The second dietary pattern in the elderly we derived in the study was the Western pattern, characterized by a higher intake of milk, other red meats, fruits, cakes and desserts and a lower intake of wheat, other cereals, and vegetables. This pattern is similar to studies elsewhere [35,36,37]. People have rapidly shifted to the Western dietary pattern both in urban and rural districts of China, especially with economic development. Milk and dairy products are a good source of protein, but there are no consistent and conclusive results with regard to the effect of milk on the lipid profile [38]. Cakes and desserts contain refined carbohydrates and added sugars, having potentially deleterious effects on metabolism by stimulating hepatic de novo lipogenesis, leading to reduced HDL-C concentrations [39]. In addition, other red meats, such as beef and mutton, have always been eaten in North China, and an Iran cohort study revealed that long-term red meat consumption has a significant positive association with the lipid profile [40]. However, fruits have been determined as healthy, with beneficial effects on HDL-C concentrations [28] and CHD [41], but other foods mentioned in this dietary pattern might reverse the plausible benefits. Thus, although no significant association was found between low HDL-C and the Western dietary pattern in the present study, our study showed a linearly increasing trend of serum TC, TG, and LDL-C levels from the lowest to the highest quartiles of this pattern. This finding is in agreement with those of other epidemiologic studies [42].

The thrifty diet might be an important special dietary pattern in the elderly population in North China. Some elders are not inclined to select nutrient-rich foods, especially in undeveloped areas [43]. This pattern includes a lot of plant-based foods, such as cereals, tubers, soybeans, vegetables, candy, and starches, but less animal-sourced foods. In the present study, this pattern was characterized by higher carbohydrate and lower dietary fat when compared with the other two dietary patterns and it had a negative relationship with the TC, LDL-C, and HDL-C concentrations. In the final model, there was a strong negative association between the thrifty pattern and HDL-C level. Recently, a strong evidence-based umbrella review identified a vegetarian diet to be associated with significantly lower concentrations of blood total cholesterol, LDL-C, and HDL-C [44]. Although the thrifty pattern is similar to the vegetarian diet or a plant-based pattern and will decrease the lipid levels, the food components in this pattern might induce malnutrition, especially among elderly Chinese. In fact, we advised the elderly people to choose abundant high-quality protein food sources, such as aquatic products, eggs, and lean meat, which will help not only reduce the blood lipid level but also maintain daily health, prevent frailty, and even prolong healthy life [45,46]. Notably, the participants in the underweight malnutrition group had lower HDL-C concentrations than those in the normal and overweight groups, indicating that nutritional improvement in elderly people benefits HDL-C concentration. Compared to the elderly people with lower BMIs, those with higher BMIs tended to select foods according to the dietary guidelines for Chinese residents and the variety of foods available to them [47]. Therefore, a gradual increase in the accessibility and variety of food is needed to promote the health status of elderly people on the thrifty dietary pattern.

The BMI is a significant indicator for several cardiovascular metabolic disorders [48,49,50]. Our study gave some meaningful results when we stratified the BMI into four quartiles. In our study, the lower quartile, the median, and the upper quartile of the BMI in the elderly were 21.6, 24.8, and 27.1 kg/m^2^. The elders in the lowest BMI subgroup tended to live on a thrifty dietary pattern, and their lipids were at a lower level. However, a negative association was observed in the balanced pattern with low HDL-C in the group with a BMI more than 27.1 kg/m^2^. In fact, the blood lipid not only indicated the possibility of cardiovascular disease but also reflected the nutritional status of the elderly people. Literature from the longevity cohort of China has reported that a higher blood lipid level is related to the health and longevity of the oldest-old population [51]. Therefore, lipid control in elderly people might need reconsideration, especially in the background of healthy aging of Chinese people. The thrifty dietary pattern would indeed decrease blood lipids, especially lowering the HDL-C levels of elderly people with low body weights. From the point of improving the nutritional status, we recommend that some healthy animal foods, such as lean meats and aquatic products, be added to their diet, apart from plant foods. However, in the elderly population with higher body weights, higher HDL-C levels were related to a balanced diet pattern. Thus, keeping relatively appropriate BMI levels by maintaining a balanced dietary pattern is recommended in the elderly.

Ineluctably, the present study had several limitations, as follows. Firstly, the cross-sectional study design made it impossible to explain the causal relationship between dietary patterns and low HDL-C. Therefore, additional prospective studies are urgently required to further confirm the findings. Secondly, we derived the dietary patterns from the food intake using factor analysis, which made the food classification, the number of factors extracted and used in the models, and dietary pattern labeling subjective to some extent [12]. Thirdly, it was inevitable to obtain different dietary patterns in variant studies owing to differences in cultures and research objectives. Therefore, it will be prudent to compare our findings with the findings of other studies. Fourthly, our study extracted three components, which could explain just 24.9% of the variance in these food items, and the 24 h dietary recall method could not represent the usual intake of the elderly participants. The final limitation was that we derived the three dietary patterns just from the elderly participants living in a specific district, and so we suggest a cautious application of this research when translated to other elderly populations. Nevertheless, this study has practical significance in demonstrating the association between dietary patterns and low HDL-C in elderly Chinese population in a geography-specific district. The present study provides a better understanding of dietary habits and the nutritional status of elderly people living in North China, which will help promote healthy aging by preventing dyslipidemia through stratification of the nutritional status strategy.

## 5. Conclusions

In this study, three main dietary patterns among elderly people living in a North China district were identified, and the association between the balanced dietary pattern, the Western dietary pattern, and the thrifty dietary pattern and low HDL-C was evaluated. The study demonstrated that the balanced dietary pattern is positively related with the HDL-C level (the HDL-C level is high), while there is a strong negative association between the thrifty dietary pattern and HDL-C (the HDL-C level is low). These findings provide a comprehensive understanding of elderly dietary patterns and are of great significance for improving diet quality, suggesting an increase in healthy animal-sourced foods for active aging of the Chinese population.

## Figures and Tables

**Table 1 nutrients-13-03308-t001:** Basic characteristics of older participants in the study living in North China.

	Normal HDL-C	Low HDL-C	*p*-Value
*N*	2646 (78.1)	741 (21.9)	
Gender (*n*, %)			<0.001
Male	1285 (75.0)	428 (25.0)	
Female	1361 (81.3)	313 (18.7)	
Age (y, mean, SD)	67.3 (5.9)	67.3 (5.9)	0.785
Age group (*n*, %)			0.758
60–64	1161 (78.0)	327 (22.0)	
65–69	772 (78.8)	208 (21.2)	
70–74	398 (78.2)	111 (21.8)	
75–79	206 (76.0)	65 (24.0)	
80–above	109 (78.4)	30 (21.6)	
Region			0.249
Urban	1105 (77.2)	327 (22.8)	
Rural	1541 (78.8)	414(21.2)	
Nationality			0.024
Han	2563 (78.4)	705 (21.6)	
Other	83 (69.7)	36 (30.3)	
Education level (*n*, %)			0.012
Primary school or below	1636 (79.3)	428 (20.7)	
Junior high school	647 (77.8)	185 (22.2)	
Senior high school or above	363 (73.9)	128 (26.1)	
Marital status (*n*, %)			0.793
Having a partner	2402 (78.1)	675 (21.9)	
Other status ^^^	244 (78.7)	66 (21.3)	
Yearly average income ^&^ (*n*, %)			0.022
Low	641 (80.6)	154 (19.4)	
Median	516 (80.4)	126 (19.6)	
High	1099 (75.8)	350 (24.2)	
No response	390 (77.8)	111 (22.2)	
Smoking ^##^			0.214
Yes	686 (76.6)	209 (23.4)	
No	1960 (78.7)	532 (21.3)	
Drinking ^&&^			<0.001
Yes	569 (83.7)	111 (16.3)	
No	2077 (76.7)	630 (23.3)	
Physical activity ^§^			0.011
Yes	567 (74.8)	191 (25.2)	
No	2079 (79.1)	550 (20.9)	
BMI (mean (SD), kg/m^2^)	24.6 (3.4)	25.8 (3.1)	<0.001
Underweight	83 (69.7)	36 (30.3)	
Normal weight	1108 (84.5)	204 (15.5)	
Overweight	1064 (74.7)	360 (25.3)	
Obesity	406 (70.5)	170 (29.5)	
Waist circumference (mean (SD), cm)	85.7 (9.6)	90.0 (8.7)	<0.001
Central obesity	1431 (74.0)	503 (26.0)	
Normal waist circumference	1215 (83.6)	238 (16.4)	
SBP (mm Hg)	146.1 (20.5)	146.2 (19.9)	0.927
DBP (mm Hg)	80.6 (10.5)	81.0 (9.9)	0.287
TC (mmol/L)	4.99 (0.88)	4.51 (0.83)	<0.001
TG (mmol/L)	1.32 (0.68)	1.97 (1.06)	<0.001
HDL-C (mmol/L)	1.38 (0.25)	0.91 (0.09)	<0.001
LDL-C (mmol/L)	3.10 (0.80)	2.95 (0.74)	<0.001

^: single or divorced or widowed. ^&^: yearly average income per capita; low is < 5000 yuan RMB; median is 5000–9999 yuan RMB; high is >= 10,000 yuan RMB. ^##^: smoking during the last 30 days. ^&&^: drinking during the last year. ^§^: physical activity of moderate and high intensity more than 150 mins per week; BMI: body mass index. SBP: systolic blood pressure; DBP: diastolic blood pressure; TC: total cholesterol; TG: triglyceride; HDL-C: high-density lipoprotein cholesterol; LDL-C: low-density lipoprotein cholesterol.

**Table 2 nutrients-13-03308-t002:** Factor loadings for dietary patterns among older participants living in North China *.

	Balanced Pattern	Western Pattern	Thrifty Pattern
Rice and products	0.29	0.21	-
Wheat and products	-	−0.56	-
Other cereals	-	-	0.60
Tubers	-	-	0.53
Dried beans	-	0.20	-
Soybean and products	-	-	0.47
Fresh vegetables	0.62	−0.20	0.24
Fresh fruits	0.52	0.38	-
Nuts	0.41	0.22	-
Pork	0.54	-	-
Other red meats	-	0.46	-
Red meat offal	-	-	-
Poultry	-	0.35	-
Milk and dairy products	-	0.53	-
Eggs	0.53	-	−0.22
Aquatic products	0.37	-	-
Cakes and desserts	-	0.38	-
Candy and starches	-	-	0.46

* factor loading ≥ |0.2|, represented a strong relation between food items and identified dietary patterns.

**Table 3 nutrients-13-03308-t003:** Characteristics of older participants according to quartiles of the three dietary patterns *.

	Balanced Pattern			Western Pattern			Thrifty Pattern		
	Q1	Q4	*p* ^§^	Q1	Q4	*p* ^§^	Q1	Q4	*p* ^§^
Age (years)	67.9 (6.1)	66.5(5.4)	<0.001	66.0 (4.9)	68.1 (6.2)	<0.001	67.8 (6.1)	66.4 (5.3)	<0.001
Female (%)	13.9	10.6	<0.001	9.4	13.1	<0.001	12.8	10.6	<0.001
Urban (%)	7.3	14.4	<0.001	8.4	13.9	<0.001	13.0	8.0	<0.001
Han nationality (%)	23.8	24.3	0.018	24.8	23.5	<0.001	24.2	24.4	0.745
Education level (high) ** (%)	1.3	5.8	0.001	2.7	5.2	<0.001	3.7	3.1	0.287
Income (high) ^#^ (%)	6.8	15.0	0.001	7.8	14.7	<0.001	11.8	7.6	<0.001
Smoking (%)	6.6	6.2	0.367	7.4	6.3	0.043	6.5	7.7	0.006
Drinking (%)	3.2	6.9	<0.001	5.3	5.1	0.725	5.4	5.6	0.404
Physical activity (%)	17.8	20.5	<0.001	19.2	20.4	0.016	19.0	20.6	0.003
BMI (kg/m^2^)	24.6 (3.4)	25.3 (3.3)	<0.001	24.7 (3.2)	25.1 (3.4)	0.005	24.9 (3.4)	24.6 (3.4)	0.017
Waist circumference (cm)	85.8 (9.8)	87.2 (9.5)	0.001	86.2 (9.6)	86.6 (9.7)	0.251	86.6 (9.7)	86.6 (9.8)	0.731
SBP (mm Hg)	146.7 (20.2)	145 (19.4)	0.220	145.6 (20)	145.2 (20.7)	0.616	145.7 (19.6)	146.2 (20.8)	0.724
DBP (mm Hg)	80.4 (10.4)	80.7 (10.6)	0.341	81.5 (10.0)	80.1 (10.5)	0.002	80.4 (10.2)	80.3 (10.7)	0.917
TC (mmol/L)	4.84 (0.93)	4.87 (0.87)	0.260	4.81 (0.87)	4.90 (0.91)	0.007	4.94 (0.89)	4.77 (0.88)	<0.001
TG (mmol/L)	1.50 (0.82)	1.45 (0.84)	0.087	1.41 (0.81)	1.48 (0.80)	0.038	1.44 (0.83)	1.49 (0.85)	0.499
HDL-C (mmol/L)	1.23 (0.29)	1.29 (0.30)	<0.001	1.28 (0.30)	1.26 (0.29)	0.073	1.28 (0.29)	1.25 (0.30)	0.029
LDL-C (mmol/L)	3.03 (0.83)	3.07 (0.77)	0.212	3.02 (0.77)	3.09 (0.82)	0.013	3.11 (0.81)	2.99 (0.78)	0.003

* values are the mean ± standard deviation for continuous variables and percentages for categorical variables; **: senior high school and above was considered as a high education level; ^#^: yearly average income more than 10,000 yuan RMB was considered as a high income; ^§^: trend from a linear regression analysis for continuous variables and Mantel-Haenszel chi-squared distribution for categorical variables.

**Table 4 nutrients-13-03308-t004:** Food and macronutrient intake in quartiles of dietary patterns among the older participants (mean, SD).

Balanced Pattern	Q1	Q2	Q3	Q4	*p* for Trend
Rice and products (g/day)	38.2 (37.6)	43.3 (37.4)	58.3 (48.7)	68.8 (59.5)	<0.001
Fresh vegetables (g/day)	125.5 (80.8)	198.5 (97.5)	258.6 (123.7)	390.7 (199.8)	<0.001
Fresh fruits (g/day)	62.0 (52.3)	84.7 (60.4)	106.5 (72.8)	187.3 (137.8)	<0.001
Nuts (g/day)	10.8 (5.6)	17.8 (11.8)	20.7 (14.6)	45.8 (42.1)	<0.001
Pork (g/day)	20.6 (15.6)	29.3 (21.0)	37.2 (25.2)	64.5 (50.6)	<0.001
Eggs (g/day)	26.1 (13.4)	37.4 (20.9)	47.8 (27.8)	64.9 (41.4)	<0.001
Aquatic products (g/day)	29.1 (22.9)	41.0 (31.9)	54.7 (52.3)	101.6 (107.3)	<0.001
Energy intake (kcal/day)	1386 (505)	1444 (498)	1572 (470)	1946 (570)	<0.001
Protein intake (g/day)	38.2 (19.1)	42.0 (16.9)	48.5 (16.0)	68.0 (25.6)	<0.001
Fat intake (g/day)	48.7 (32.9)	50.8 (28.1)	57.8 (29.9)	79.3 (34.9)	<0.001
Carbohydrate intake (g/day)	207.4 (95.0)	213.7 (89.8)	224.8 (89.2)	256.2 (97.0)	<0.001
**Western Pattern**					
Rice and products (g/day)	41.6 (33.1)	47.2 (38.5)	61.9 (58.1)	59.7 (54.3)	<0.001
Wheat and products (g/day)	299.6 (97.1)	196.8 (66.7)	151.3 (69.1)	138.2 (74.3)	<0.001
Dried beans (g/day)	15.5 (15.5)	20.7 (19.4)	27.5 (27.8)	39.3 (42.2)	<0.001
Fresh vegetables (g/day)	322.8 (192.3)	228.7 (141.1)	205.6 (140.1)	216.4 (154.9)	<0.001
Fresh fruits (g/day)	80.6 (56.5)	98.1 (78.4)	115.2 (84.8)	176.0 (139.7)	<0.001
Nuts (g/day)	24.9 (23.4)	28.1 (23.5)	24.3 (25.5)	44.5 (43.5)	<0.001
Other red meats (g/day)	26.5 (16.7)	32.9 (21.9)	34.7 (21.7)	74.3 (70.7)	<0.001
Poultry (g/day)	33.7 (30.9)	35.5 (24.4)	37.4 (21.6)	61.8 (45.6)	<0.001
Milk and dairy products (g/day)	72.1 (24.5)	80.1 (37.5)	101.8 (50.5)	172.8 (98.4)	<0.001
Cakes and desserts (g/day)	22.1 (20.1)	34.5 (24.7)	35.4 (21.6)	63.5 (62.0)	<0.001
Energy intake (kcal/day)	1780 (541)	1472 (481)	1441 (534)	1655 (594)	<0.001
Protein intake (g/day)	55.1 (22.2)	43.5 (17.2)	41.9 (19.0)	56.1 (27.7)	0.689
Fat intake (g/day)	60.7 (33.7)	53.5 (34.1)	55.0 (30.0)	67.3 (35.4)	<0.001
Carbohydrate intake (g/day)	266.3 (94.9)	213.2 (87.4)	203.7 (89.3)	219.0 (94.4)	<0.001
**Thrifty Pattern**					
Other cereals (g/day)	24.4 (20.2)	42.9 (29.3)	62.2 (42.0)	123.3 (90.2)	<0.001
Tubers (g/day)	46.6 (27.9)	61.7 (37.6)	88.1 (56.5)	159.1 (118.5)	<0.001
Soybean and products (g/day)	7.1 (5.3)	10.1 (7.8)	16.5 (13.9)	28.5 (27.2)	<0.001
Fresh vegetables (g/day)	180.7 (118.7)	227.1 (144.9)	274.6 (179.5)	290.9 (185.7)	<0.001
Eggs (g/day)	59.8 (39.7)	46.0 (29.7)	43.4 (28.2)	42.5 (30.3)	<0.001
Candy and starches (g/day)	5.6 (5.8)	10.1 (8.1)	16.6 (13.7)	31.9 (30.3)	<0.001
Energy intake (kcal/day)	1443 (494)	1399 (492)	1600 (508)	1906 (581)	<0.001
Protein intake (g/day)	47.5 (23.9)	42.2 (18.4)	49.0 (22.8)	57.9 (23.2)	<0.001
Fat intake (g/day)	61.9 (32.1)	54.8 (33.7)	60.6 (34.2)	59.4 (34.7)	0.768
Carbohydrate intake (g/day)	181.3 (72.8)	193.1 (72.3)	226.9 (79.7)	300.9 (101.4)	<0.001

**Table 5 nutrients-13-03308-t005:** Association between low HDL-C and quartiles of dietary patterns in the older participants by BMI subgroups (BMI < 21.6 kg/m^2^).

	Q1	Q2		Q3		Q4		*p* for Trend
		OR	95% CI	OR	95% CI	OR	95% CI	
**Balanced Pattern**								
Crude model	1.00	0.56	0.21–1.45	0.52	0.19–1.42	0.32	0.12–0.88	0.174
Model 1 *	1.00	0.60	0.25–1.46	0.46	0.18–1.24	0.36	0.12–1.07	0.256
Model 2 ^&^	1.00	0.61	0.24–1.54	0.48	0.17–1.32	0.36	0.12–1.10	0310
Model 3 ^a^	1.00	0.58	0.23–1.43	0.43	0.15–1.22	0.34	0.10–1.09	0.275
Model 3+WC+BP	1.00	0.60	0.25–1.44	0.47	0.16–1.36	0.32	0.10–1.12	0.319
**Western Pattern**								
Crude model	1.00	1.45	0.58–3.65	0.75	0.31–1.82	0.65	0.24–1.82	0.448
Model 1 *	1.00	1.15	0.52–2.56	0.40	0.15–1.08	0.45	0.16–1.24	0.183
Model 2 ^&^	1.00	1.19	0.55–2.57	0.41	0.15–1.12	0.46	0.16–1.31	0.179
Model 3 ^b^	1.00	0.99	0.46–2.13	0.33	0.12–0.94	0.42	0.15–1.18	0.101
Model 3+WC+BP	1.00	1.14	0.50–2.62	0.44	0.15–1.24	0.57	0.20–1.62	0.199
**Thrifty Pattern**								
Crude model	1.00	4.37	1.64–11.68	4.49	1.52–13.32	3.38	1.34–8.53	0.013
Model 1 *	1.00	2.90	1.21–6.96	2.63	0.99–6.98	2.54	1.01–6.41	0.076
Model 2 ^&^	1.00	3.01	1.19–7.60	2.86	1.16–7.05	2.69	1.06–6.86	0.057
Model 3 ^c^	1.00	3.64	1.34–9.87	3.79	1.37–10.52	3.28	1.17–9.18	0.037
Model 3+WC+BP	1.00	3.95	1.37–11.38	4.75	1.52–14.80	3.31	1.05–10.40	0.044

Model 1 *: adjusted for district (urban/rural), age groups (60 y-, 65 y-, 70 y-, 75 y-, 80 y-), gender (male/female), marital status (having a partner/other status), nationality (Han/others), income (low/medium/high/no response), and education level (low/middle/high); Model 2 ^&^: adjusted for smoking (yes/no), drinking (yes/on), physical activity (yes/no), and Model 1; Model 3 ^a^: adjusted for the Western dietary pattern, the thrifty pattern, energy intake, and model 2; Model 3 ^b^: adjusted for the Western pattern, the thrifty dietary pattern, energy intake, and Model 2; Model 3 ^c^: adjusted for the balanced pattern, the Western dietary pattern, energy intake, and Model 2.

**Table 6 nutrients-13-03308-t006:** Association between low HDL-C and quartiles of dietary patterns in the older participants by BMI subgroups (21.6 <= BMI < 24.8 kg/m^2^).

	Q1	Q2		Q3		Q4		*p* for Trend
		OR	95% CI	OR	95% CI	OR	95% CI	
**Balanced Pattern**								
Crude model	1.00	0.71	0.31–1.60	0.61	0.24–1.57	0.49	0.23–1.05	0.314
Model 1 *	1.00	0.68	0.29–1.59	0.55	0.22–1.40	0.37	0.16–0.83	0.104
Model 2 ^&^	1.00	0.74	0.31–1.75	0.65	0.26–1.65	0.51	0.22–1.17	0.441
Model 3 ^a^	1.00	0.66	0.29–1.53	0.64	0.25–1.64	0.46	0.20–1.06	0.337
Model 3+WC+BP	1.00	0.61	0.28–1.29	0.61	0.25–1.47	0.46	0.19–1.09	0.332
**Western Pattern**								
Crude model	1.00	1.19	0.52–2.72	1.14	0.48–2.68	0.61	0.28–1.32	0.369
Model 1 *	1.00	1.34	0.64–2.82	1.02	0.46–2.29	0.50	0.24–1.06	0.060
Model 2 ^&^	1.00	1.38	0.66–2.87	1.16	0.52–2.60	0.44	0.21–0.93	0.018
Model 3 ^b^	1.00	1.24	0.57–2.68	0.96	0.41–2.23	0.39	0.18–0.85	0.017
Model 3+WC+BP	1.00	1.11	0.50–2.47	0.90	0.38–2.15	0.37	0.17–0.82	0.018
**Thrifty Pattern**								
Crude model	1.00	1.09	0.42–2.83	1.04	0.46–2.38	1.36	0.68–2.75	0.829
Model 1 *	1.00	0.97	0.39–2.42	1.07	0.45–2.55	1.55	0.71–3.39	0.564
Model 2 ^&^	1.00	0.86	0.34–2.13	1.04	0.42–2.56	1.61	0.74–3.53	0.375
Model 3 ^c^	1.00	0.90	0.36–2.27	1.26	0.52–3.05	2.00	0.88–4.56	0.247
Model 3+WC+BP	1.00	0.88	0.35–2.18	1.20	0.51–2.85	2.15	0.90–5.12	0.225

Model 1 *: adjusted for district (urban/rural), age groups (60 y-, 65 y-, 70 y-, 75 y-, 80 y-), gender(male/female), marital status (having a partner/other status), nationality (Han/others), income (low/medium/high/no response), and education level (low/middle/high); Model 2 ^&^: adjusted for smoking (yes/no), drinking (yes/on), physical activity (yes/no), and Model 1; Model 3 ^a^: adjusted for the Western dietary pattern, the thrifty pattern, energy intake, and Model 2; Model 3 ^b^: adjusted for the Western pattern, the thrifty dietary pattern, energy intake, and model 2; Model 3 ^c^: adjusted for the balanced pattern, the Western dietary pattern, energy intake, and Model 2.

**Table 7 nutrients-13-03308-t007:** Association between low HDL-C and quartiles of dietary patterns in the older participants by BMI subgroups (24.8 <= BMI < 27.1 kg/m^2^).

	Q1	Q2		Q3		Q4		*p* for Trend
		OR	95% CI	OR	95% CI	OR	95% CI	
**Balanced Pattern**								
Crude model	1.00	0.75	0.36–1.58	0.61	0.28–1.36	1.05	0.53–2.07	0.520
Model 1 *	1.00	0.76	0.36–1.58	0.64	0.28–1.51	1.06	0.53–2.08	0.568
Model 2 ^&^	1.00	0.80	0.38–1.65	0.63	0.26–1.48	1.38	0.69–2.76	0.215
Model 3 ^a^	1.00	0.81	0.38–1.69	0.64	0.26–1.57	1.50	0.64–3.50	0.154
Model 3+WC+BP	1.00	0.75	0.36–1.55	0.63	0.26–1.51	1.47	0.61–3.57	0.131
**Western Pattern**								
Crude model	1.00	0.70	0.31–1.61	0.90	0.45–1.83	0.78	0.37–1.65	0.829
Model 1 *	1.00	0.80	0.39–1.63	1.04	0.52–2.05	0.85	0.43–1.69	0.863
Model 2 ^&^	1.00	0.85	0.40–1.83	0.98	0.48–1.97	0.81	0.40–1.65	0.914
Model 3 ^b^	1.00	0.86	0.40–1.85	0.99	0.50–1.98	0.82	0.41–1.64	0.924
Model 3+WC+BP	1.00	0.98	0.44–2.19	1.06	0.52–2.15	0.90	0.44–1.82	0.974
**Thrifty Pattern**								
Crude model	1.00	1.90	0.90–4.01	0.89	0.44–1.77	1.75	0.93–3.31	0.064
Model 1 *	1.00	1.80	0.90–3.62	0.73	0.38–1.41	1.50	0.79–2.85	0.054
Model 2 ^&^	1.00	1.67	0.83–3.35	0.72	0.37–1.39	1.52	0.79–2.91	0.078
Model 3 ^c^	1.00	1.64	0.82–3.31	0.71	0.34–1.46	1.52	0.74–3.10	0.077
Model 3+WC+BP	1.00	1.87	0.88–3.96	0.85	0.40–1.80	1.69	0.82–3.49	0.085

Model 1 *: adjusted for district (urban/rural), age groups (60 y-, 65 y-, 70 y-, 75 y-, 80 y-), gender (male/female), marital status (having a partner/other status), nationality (Han/others), income (low/medium/high/no response), and education level (low/middle/high); Model 2 ^&^: adjusted for smoking (yes/no), drinking (yes/on), physical activity (yes/no), and Model 1; Model 3 ^a^: adjusted for the Western dietary pattern, the thrifty pattern, energy intake, and Model 2; Model 3 ^b^: adjusted for the Western pattern, the thrifty dietary pattern, energy intake, and Model 2; Model 3 ^c^: adjusted for the balanced pattern, the Western dietary pattern, energy intake, and Model 2.

**Table 8 nutrients-13-03308-t008:** Association between low HDL-C and quartiles of dietary patterns in the older participants by BMI subgroups (BMI >= 27.1 kg/m^2^).

	Q1	Q2		Q3		Q4		*p* for Trend
		OR	95% CI	OR	95% CI	OR	95% CI	
**Balanced Pattern**								
Crude model	1.00	0.64	0.30–1.33	0.33	0.16–0.66	0.62	0.30–1.26	0.019
Model 1 *	1.00	0.56	0.27–1.17	0.25	0.12–0.52	0.36	0.17–0.78	0.003
Model 2 ^&^	1.00	0.64	0.30–1.37	0.30	0.14–0.63	0.40	0.18–0.87	0.012
Model 3 ^a^	1.00	0.61	0.28–1.30	0.29	0.14–0.63	0.39	0.17–0.90	0.014
Model 3+WC+BP	1.00	0.60	0.27–1.31	0.29	0.13–0.63	0.38	0.16–0.88	0.013
**Western Pattern**								
Crude model	1.00	1.21	0.59–2.50	1.38	0.66–2.89	1.44	0.70–2.94	0.767
Model 1 *	1.00	1.09	0.54–2.23	1.17	0.60–2.27	1.10	0.57–2.14	0.974
Model 2 ^&^	1.00	1.16	0.57–2.36	1.32	0.68–2.57	1.30	0.66–2.55	0.833
Model 3 ^b^	1.00	1.07	0.52–2.22	1.16	0.59–2.28	1.28	0.64–2.58	0.906
Model 3+WC+BP	1.00	0.99	0.47–2.08	1.07	0.54–2.13	1.20	0.60–2.39	0.948
**Thrifty Pattern**								
Crude model	1.00	1.05	0.53–2.08	1.04	0.51–2.12	1.48	0.77–2.88	0.624
Model 1 *	1.00	1.02	0.51–2.06	0.95	0.48–1.86	1.54	0.82–2.90	0.451
Model 2 ^&^	1.00	1.04	0.51–2.10	1.01	0.51–2.00	1.72	0.92–3.23	0.286
Model 3 ^c^	1.00	1.08	0.53–2.20	1.11	0.56–2.20	1.98	1.00–3.91	0.206
Model 3+WC+BP	1.00	1.07	0.52–2.20	1.10	0.54–2.21	2.03	1.02–4.05	0.180

Model 1 *: adjusted for district (urban/rural), age groups (60 y-, 65 y-, 70 y-, 75 y-, 80 y-), gender (male/female), marital status (having a partner/other status), nationality (Han/others), income (low/medium/high/no response), and education level (low/middle/high); Model 2 ^&^: adjusted for smoking (yes/no), drinking (yes/on), physical activity (yes/no), and Model 1; Model 3 ^a^: adjusted for the Western dietary pattern, the thrifty pattern, energy intake, and Model 2; Model 3 ^b^: adjusted for the Western pattern, the thrifty dietary pattern, energy intake, and Model 2; Model 3 ^c^: adjusted for the balanced pattern, the Western dietary pattern, energy intake, and Model 2.

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
