# Peer review of "Association between Dietary Patterns and Low HDL-C among Community-Dwelling Elders in North China"

_nutrients, 2021, doi:10.3390/nu13103308_

Round 1
Reviewer 1 Report
The manuscript is original and of importance in its field. The issue is interesting due to the role of dietary pattern in CVD risk. Introduction section has been adequately presented. The scientific rationale and the conceptual design are of relevance, and the article was conceived correctly. Moreover, methods are not clearly mentioned. Language should be corrected by a native English person. Statistical analysis should be extended.
Some limitations are highlighted. The authors can be trusted to make these:
- Methods are not clearly mentioned.
- The groups of Table 2 should be better described in Material Section. The Authors should describe the different type of dietary pattern.
- Description of data collection on physical activity should added in Material Section
- Please provide a statistical analysis of the data correlation among the most significant parameters.
- As further statistical analysis, the authors should divide the groups according to the to the BMI. The results and the discussion should be reviewed based on the new observations.
- In the Conclusion Section, it would be much better in my opinion if the Authors give a recommendation of diet which is good for the prevention of malnutrition and dyslipidemia, as the point of view of the nutritionist.
- The limits of the research should be added and discussed.
- Recent references should be included to strengthen the argument.
- The paper is clearly written, although the language should be corrected by a native English person.
- The manuscript could be published in Nutrients after minor revisions.
Author Response
Thanks for your kindly comments for the manuscript. A major revision in the method, further statistical analysis, relevant results, discussion, conclusion and reference had been conducted according to your professional suggestion.
Just as you have suggested for the method and statistical analysis, physical activity data collection was added in method section, and food items was described in detail according to 18 food categories in Table 2. For the dietary pattern we derived in the article, we described the difference of them and add a new table to describe the characteristics of participants (contained age, gender, BMI, blood lipids profile, etc.) by each dietary pattern. Pearson correlation analysis among the most significant parameters such as age, BMI, blood lipids and physical activity time was added. BMI was a significant indicator for many diseases especially in the elders, BMI distribution in the elderly was not similar as other age groups, so we stratified BMI for four groups by its quartiles, and some new result was received and we explained in the discussion section correspondingly.
Furthermore, we supplied some recommendation of diet and nutrition to prevent dyslipidemia for different nutritional status of the elderly. The limitation of the manuscript was discussed and recent reference was added to support the argument.
Thanks again and best wishes for you.
Reviewer 2 Report
The paper reports the results of a cross-sectional investigation (China Adults Chronic Diseases and Nutrition Surveillance in 2015) performed on 3387 elderly people living in North-China, focused on the association between low HDL-C and dietary patterns. Lifestyle and health characteristics and food intakes have been collected, together with anthropometry, blood pressure and fasting serum lipids. Three dietary patters were identified: balanced, western and thrifty patterns, and the risk calculated for Q4 of adherence in comparison with Q1. After adjusting for potential confounding factors, the risk for low HDL-C decreased in the participants with higher scores in balanced dietary patterns (OR=0.58, 95%CI: 0.36-0.93, P=0.010), while increased in participants with greater adherence to thrifty dietary pattern (OR=2.11, 95%CI: 1.40-3.18, P=0.003). No statistically significant association was found between western dietary pattern and low HDL-C.
Nevertheless, the thrifty pattern seems to be a plant based pattern, and it is well described that people following a plant-based diet have lower TC levels and lower HDL-C levels, but also a reduced CV risk.
The importance of HDL-C levels as a risk factor is controversial, as they depend on LDL-C levels, whose role in CV risk is, on the contrary, well documented.
This aspect is not sufficiently featured in the paper, also in consideration that Table 3 shows for the balanced pattern a non-significant increase and for the thrifty pattern a significant reduction of TC and LDL-C. Why was this aspect not evaluated and appropriately discussed? I trust that it should be accurately analyzed, as it can influence HDL levels.
I suggest finding out more inspiration on the issue from the umbrella review by Oussalah, where this aspect is well discussed.
Oussalah A, Levy J, Berthezène C, Alpers DH, Guéant JL. Health outcomes associated with vegetarian diets: An umbrella review of systematic reviews and meta-analyses. Clin Nutr. 2020 Nov;39(11):3283-3307. doi: 10.1016/j.clnu.2020.02.037. Epub 2020 Mar 11. PMID: 32204974.
I have no other observation or suggestion, except for some mistakes in the orthography and some missing verbs.
Author Response
Thank you for your serious review of the article. I have read the literature you provided and I think this article is very instructive for me to discuss dyslipidemia. I quite agree with your opinion that there is a negative relationship between plant-based dietary pattern and blood lipid level, which is supported by sufficient literatures. The reason we choose the topic of HDL-C and diet objectively based on the fact of the special blood lipids profile of China when compared with western countries. The typical dyslipidemia in China is low HDL-C and hypertriglyceridemia, but not hypercholesterolemia and high LDL-C. HDL-C is related with genetic factors, but it is also related to the one's nutritional status. The proportion of patients with dyslipidemia treated with statins in China is very low and specific drug for increasing HDL-C had no sufficient evidence, so we try to analysis the influencing factors from the nutrition point.
Generally, total lipid levels, dietary intake and BMI will be decreased with increase of age, but HDL-c will rise in old life. Under the background of aging population in our country, we aimed to study the relationship between low HDL-c and diet, and to explore how dietary improvement can balance overall lipid levels in the elderly population. This paper is not to put forward dietary recommendations to control dyslipidemia strictly, but from the perspective to improve healthy aging. Because the blood lipid level of the elderly reflects the nutritional status of the body to a certain extent, and literatures from the longevity cohort of our country also reported that higher blood lipid level was related to the health and longevity of the elderly. Therefore, balanced diet that includes not only plant-based foods but also moderate animal foods will be beneficial for the nutritional status of older adults. We do not particularly recommend vegetarian diet for the underweight elderly, because vegetarian diet contains insufficient polyunsaturated fatty acids and nearly non-high quality protein from animal foods. In order to maintain an appropriate level of blood lipid, a balanced diet based plant food and healthy animal food as the supplement is needed.
we made a major revision for your suggestion and added some statistical analysis. A modified conclusion was as follows: for the elderly population with low body weight, the thrifty pattern was negatively related with HDL level, so it is not recommended them to choose this pattern. While in the elderly population with higher body weight, they should continue to maintain a balanced dietary pattern but keeping a moderate BMI.
Round 2
Reviewer 2 Report
Authors addressed my suggestion.